# Translation, Cross-Cultural Adaptation, and Validation of the Foot Posture Index (FPI-6)—Italian Version

**DOI:** 10.3390/healthcare11091325

**Published:** 2023-05-05

**Authors:** Serena Loreti, Anna Berardi, Giovanni Galeoto

**Affiliations:** 1Department of Anatomical, Histological, Forensic and Orthopedic Sciences, Sapienza University of Rome, Piazzale Aldo Moro 5, 00185 Rome, Italy; 2Department of Human Neurosciences, Sapienza University of Rome, Viale dell’Università 30, 00185 Rome, Italy; 3IRCSS Neuromed, Via Atinense 18, 86077 Pozzilli, Italy

**Keywords:** podology, assessment, validation, foot

## Abstract

Since foot posture is one of the main predictors of lower limb musculoskeletal injuries, it is crucial to use appropriate tools to define the foot’s posture. The Foot Posture Index is, therefore, a reliable method to measure foot posture and is widely known and used in clinics and research. This study aimed to translate the Foot Posture Index 6 (FPI6) into Italian and to assess its psychometric properties. Translation and cross-cultural adaptation were obtained using a popular guideline. Two examinators assessed 68 subjects, and data were collected to test intra/inter-rater reliability, internal consistency and cross-cultural validity. The Italian version of FPI6 showed excellent inter- and intra-rater reliability (ICC 0.96 and 0.97), and Cronbach’s alpha coefficient was 0.9, thus showing excellent internal consistency. The FPI-6 version has proved to be reliable in terms of inter- and intra-rater reliability and can, therefore, be used in clinical practice and scientific research.

## 1. Introduction

The foot is a complex structure with many articulations and degrees of freedom that play an essential role in static posture and dynamic activities [1]. Variations in foot posture, such as pes planus (low-arched foot) or pes cavus (high-arched foot), are thought to represent an intrinsic risk factor for injury due to altered movement of the lower extremity [2,3]. The most common problem associated with pes planus is excessive pronation during weight-bearing activities, leading to impaired plantar load distribution, excessive stress in the foot, ankle and knee joints, and compensatory internal rotation of the hip joint [4]. Pes cavus is reported to have less mobility and be related to injury, leading to reduced shock attenuation or increased peak plantar pressures [5]. Some authors suggest a link between foot posture and knee osteoarthritis (OA) [6,7], lower back pain [8,9,10], and postural instability [11]. Static foot assessment is commonly performed in clinical practice to classify foot type, identify possible injury-related etiological factors, and prescribe therapeutic interventions [12]. Applying a valid and reliable system of foot type classification becomes essential to evaluate any proposed relationship between foot structure and function and between foot shape and risk of injury. Different methods of foot type classification have often been used, making the comparison of the results and drawing sound conclusions impossible [13].

The Foot Posture Index-6 (FPI-6) is a simple and reliable quantification tool to assess static foot alignment in all three planes [14]. The Foot Posture Index comprises six individual parameters, to which a score from −2 to +2 can be allocated. Negative numbers indicate supination, and positive numbers indicate pronation. The total result defines the foot posture and ranges between −12 and +12.

Results equal to or higher than 10 define a highly pronated foot, +9 to +6 pronated, +5 to 0 average, −1 to −4 supinated, and ≤5 highly supinated [14,15,16]. The patient is placed in a relaxed standing position and evaluation starts with palpation of the head of the talus. Five visual evaluations follow: the above and below malleolar curves, the position of the calcaneus on the frontal plane, bulging in the talonavicular region, and, finally, the height and congruence of the medial longitudinal arch.

The scale was validated in different countries: U.S.A., Australia, Great Britain, Thailand, Brazil, Japan, China. This validation included different population health individuals, foot/ankle injury, indoor football players, patella femoral pain syndrome, knee osteoarthritis, low back pain, and pediatric and elderly populations. Moreover, previous studies showed optimal psychometric properties with Cronbach’s alpha 0.83 and test-retest (intra and inter-operator), ranging between 0.69–0.99.

Normative values are available [15,17,18]. In the study of Redmond et al., no difference between the FPI scores of males and females and between BMIs were found. Systematic differences from the normal adults were confirmed in patients with neurogenic and idiopathic causes [15]. Rokkedal-Lausch et al., in 2013, reported no significant differences in FPI scores between individuals with right, left, or no limb dominance [15]. Gijon-Nogueron et al., in 2016, evaluated normative values in the pediatric population that produced mean values of 3.74 (SD 2.93) points for the right foot and 3.83 (SD 2.92) for the left. The 50th percentile was 4 points for both genders and feet, except for the right foot among the girls, which was slightly lower, at 3 points. The 85th percentile, which is considered to represent the boundary between the normal and the pronated foot among children, was 6 points, uniformly among the subjects [18].

The foot posture index was subject to a rash analysis, which demonstrated the validity and reliability of the tool, which was found to be suitable for a range of clinical applications and to yield high-quality linear metric data [17]. Over the years, the FPI has been widely used in clinics and research [12,16,18,19,20,21,22].

The FPI-6 has always been used in its original language or a non-official translated version in Italy. Therefore, this study aimed to translate, culturally adapt, and measure the psychometric properties of the Italian version of the Foot Posture Index.

## 2. Methods

### 2.1. Translation and Cultural Adaptation Procedure

The translation of the Foot Posture Index and its use manual in Italian was carried out following the “Principles of Good Practice for the Translation and Cultural Adaptation Process for Patient-Reported Outcomes (PRO) Measures: Report of the ISPOR Task Force for Translation and Cultural Adaptation” guidelines [23].

Three individual translations were obtained, respectively, from three native Italian speakers, all healthcare professionals (V^1^)(V^2^)(V^3^). The three translations were then agreed during a meeting, resulting in only one document (V^4^). This version was then re-translated into its original language (V^5^) by a native speaker of English, a non-healthcare professional, and shared with the lead author of the original article. Once the re-translation was approved, version (V^5^) was created by adjusting version (V^4^), following the author’s advice. Version (V^5^) was then submitted to a group of non-healthcare professionals to evaluate its readability, simplicity, and fluidity of language and logical sense of translation. The text was modified according to the suggestions and the version (V^6^) was obtained. V^6^ was pre-tested by five podiatrists to identify any potential misinterpretation problems that should be disambiguated. The pre-testers completed all assessment items, and no section was found to be confusing or ambiguous, so there was no need to modify V^6^ according to the suggestions provided by the pre-testers.

The final version (V^7^) of the User Manual and Guide and Data Sheet of the FPI was produced.

### 2.2. Validation Procedure

In a 2008 article, Dr. Redmond [15] determined the normative values for the Foot Posture Index by establishing that very young and elderly patients have a greater tendency to pronate.

The same study demonstrated that BMI does not affect foot posture and that there is also no gender-specific trend.

For this reason, we decided to include 68 volunteers of both sexes, ranging between 18 to 86 years of age (mean: 47.1 ± 15.5) and body mass index (BMI) between 17.6 to 43.7 kg/m^2^ (mean: 25.3 ± 5.3). Being able to remain in the orthostatic position during the assessments was the inclusion criterion.

All participants signed an informed consent form.

Exclusion criteria were represented by musculoskeletal injuries to the lower limbs in the previous three months, foot deformities, and pain.

### 2.3. Assessment Procedure and Reliability

The FPI6 Italian version was assessed independently by two physicians, i.e., physician 1 and physician 2, to assess intra-operator reproducibility.

Physician 1 was a podiatrist with a 10-year experience and knowledge of FPI, whereas the second doctor only had one-year experience and did not know or ever used the FPI6. Foot posture assessment was performed via visual appraisal and clinical evaluation: there needs to be contact between the operator and the patient for only one item (Talar palpation).

Physicians assessed the subject’s dominant legs using the Italian guide developed for this study. The two assessed the FPI6 about 5 min from each other. The scale was again assessed between days 2 and 10 to the same population by physician 1 to examine inter-rater reproducibility. Participants were asked to stand up in a relaxed position and choose the support base’s width and angle after taking a few steps on the spot. They were instructed to stand still during the assessment and look straight with their arms by their side and were also advised not to turn around during the exam and not to compromise observations.

Physician 1 and physician 2 assessed the FPI walking around the participant following the datasheet. During the Physician 1 assessment, the other physician was in another room and vice versa; immediately after the first evaluation, the second rater performed the assessment [24].

We chose to conduct the two assessments using the same position to avoid any patient positioning that could alter the foot posture.

All items were subject to assessment, i.e., talar head palpation (1), curves above and below the lateral malleolus (2), inversion/eversion of the calcaneus (3), prominence in the region of the TNJ (4), congruence of the medial longitudinal arch (5), and abd/adduction forefoot on rearfoot (6).

The scores for each item ranged between −2 to +2, which gave a total score corresponding to the FPI6 of that foot. All data were recorded [25].

### 2.4. Statistical Analysis

Consistent with the “COnsensus-based Standards for the selection of health Measurement INstruments” checklist recommendations (COSMIN), we assessed the reliability and validity of the FPI6 as follows: Cronbach’s alpha for internal consistency needed to be >0.7 to establish the degree of agreement between the various items; the ICC test–retest reliability needed to be >0.7 to establish the stability of the individual measurements carried out at different times; and Pearson’s correlation coefficient between FPI6 and demographic characteristics of the population for Cross-cultural validity. The Pearson correlation coefficient can be interpreted as follows: 0 indicates no linear relationship; +1/−1 indicates a perfect linear positive/negative relationship; values between 0 and 0.3 (0 and −0.3) indicate a weak linear positive (negative) relationship through a shaky linear rule; values ranging from 0.3 to 0.7 (−0.3 and −0.7) indicate a moderate positive (negative) linear relationship through a fuzzy-firm linear rule; and values between 0.7 and 1.0 (−0.7 and −1.0) indicate a strongly positive (negative) linear relationship through a firm linear rule. The significance level was set for *p*-value less than or equal to 0.05. All statistical analyses were performed using IBM-SPSS version 23.00. We certify that all applicable institutional and governmental regulations concerning the ethical use of human volunteers/animals were followed during this research [26].

## 3. Results

### 3.1. Translation

During the translation process of the User’s Guide and Manual, some words and concepts proved to be difficult to understand because of their literal translation, which did not share the same conceptual meaning as in Italian: item (a); measures (b); user (c); observation (d); neutral calcaneal stance position (e). Translation divergences and final solutions chosen by the researcher are shown in Table 1. The researchers interpreted and modified these words, then applied them throughout the original paper. The final version of the scale is presented in Table 2.

### 3.2. Reliability

Participants included 68 adults, 37 females, and 31 males, aged 18 to 86 years. The participants’ demographic and clinical characteristics are shown in Table 3.

Test–retest and intra–rater reliability [26]: Thirty out of sixty-eight included patients were submitted to test–retest and intra–rater reliability procedures (mean age 47.1 ± 15.5 years), i.e., 31 males, 37 females. The test-retest reliability for each item is reported.

The Italian version of the FPI-6 showed excellent inter- and intra-operator reliability (ICC = 0.96 and 0.97) for the dominant lower limb (Table 4).

Only Item 4 of the FPI6 (bulging in the region of the talonavicular joint) showed adequate inter- and intra-rater reliability (ICC = 0.74 and 0.66, respectively). All other items displayed excellent reliability (ICC ≥ 0.75).

Internal consistency: The internal consistency was calculated on all 68 included cases. The Cronbach’s alpha value was 0.90. The item-total correlation value showed that the item’s association with the total score was excellent: five out of six items obtained a score > 0.70. Item 5 displayed a score of 0.55. From the “Alpha if Deleted” evaluation, we can assert that all items contribute to the evaluation of the scale construct because, if any of the items were removed, the total Cronbach’s alpha would not increase (Table 5).

Cross-cultural validity: Pearson’s correlation coefficient showed no statistically significant correlations between the FPI6 and the population’s demographic characteristics.

We also tested the correlation between weight, height, BMI, and total score of the FPI6 and found no correlation (Table 6).

## 4. Discussion

In a static position, observation of the loaded foot is a recurrent examination in medical offices and healthcare settings. The result of this observation is often unquantified, except when using generic terms, such as “flat foot” and “cavus foot”, or when adding adjectives, such as “severe” or “mild”, to describe the extent of the observed characteristics [27,28].

This study was designed to introduce a very useful tool, the Foot Posture Index (FPI) [14] in its Italian version into Italian clinical practice and research. The FPI is a tool created to quantify foot posture in a static position. It is a quick, easy, reliable measurement method that minimizes the subjectivity of clinical evaluation methods. In the original article of the method description, the authors reviewed 119 articles and identified 36 clinical characteristics from which they then developed a tool for evaluating the three fundamental districts of the foot (forefoot, midfoot, and hindfoot) and their position in relation to the three planes of the body. The FPI was then repeatedly validated by comparing it with other evaluation methods [13,14,24,25,29]. The datasheet and User Manual and Guide have been translated and validated into Italian.

The translation and cultural adaptation were performed with the help of the “Translation and Cultural Adaptation of Patient Reported Outcomes Measures—Principles of Good Practice guidelines” [23] and with the involvement of the original authors and a group of experts, who have been committed to ensuring the original meaning of the manual is kept. An amount of 68 volunteers, aged 18 to 86 were recruited. These included 31 males and 37 females who did not have pain, foot or leg deformities, or disability.

The population sample of previous validations available in the literature was highly variable in terms of the subjects’ number and age [14,28,30,31,32,33,34].

In a 2008 article [15], Dr Redmond and other authors determined a correlation between age and the FPI, for which children and the elderly had a greater tendency to a pronated foot posture.

However, in this study, authors decided to include subjects from 18 years of age with no maximum age limit because, as reported in the literature, the elderly are part of the population to which the FPI was addressed [15].

The tool developers decided not to include minors under 18 because of their peculiar physiological time-related changes. The child’s foot was considered a topic that needed to be treated more specifically and independently.

The Redmond article determined no relationship between BMI and foot posture. Possible correlations between BMI and excess pronation were also investigated with the data obtained in the present study. As in Dr. Redmond’s study, no correlation emerged in this sense.

On the other hand, it was impossible to draw any statistical considerations or evaluations regarding the age-dependent correlation due to the small sample of subjects over 65.

The FPI was assessed, as indicated in the User Guide and Manual in the Italian version. It was performed only on the foot the tested subject reported as predominant. The overall individual assessment results were used for statistical analysis, but considerations were also drawn from single items. In the FPI-6 original version, the instrument’s reliability was evaluated by three examiners, with different levels of experience in clinical settings (no experience, nine-year experience, and 30-year experience) [35].

For this study, we chose two examiners with different levels of knowledge of the instrument and experience: podiatrists, one with ten-year experience and one with only one-year experience. The aim was to test the tool’s validity regardless of the operator’s degree of experience. The tool provided excellent results in terms of intra- and inter-operator reliability. Reliability between raters was excellent (ICC 0.97) and intra-raters (ICC 0.96), thus demonstrating that the instrument is reliable even with little practice and experience. However, in the user manual, the author recommends at least thirty practice tests before using the tool in clinical practice.

The Cronbach’s Alpha coefficient demonstrated excellent internal consistency (0.90), and the “Alpha deleted” confirms that all the items contributed to the evaluation of the scale construct with values consistently higher than 0.8. All the items demonstrated an ICC greater than 0.65, although the lowest value in both the inter-operator (0.74) and intra-operator (0.66) scores was item 4, i.e., “protrusion of the talonavicular joint”. This is believed to be due to the operators’ uncertainty concerning the fact that the photo in the user manual, which highlighted the area to be taken into consideration during the item observation, did not seem to correspond to that of interest for the joint. Overall, the ICC’s value for the total scores showed 0.96 and 0.97 for intra and inter-rater reliability, respectively.

In conclusion, evidence-based medicine needs reliable tools to quantify symptoms and observations. Even though there is a lack of subgroup analysis due to the relatively small sample size, thanks to this study, the Italian version of the FPI6 has proven to be a reliable, valid, easy-to-understand-and-use tool, helpful in assessing the foot posture in a static position. This study provides Italian professionals of any branch with a tool for evaluation and interdisciplinary communication. Moreover, now it is possible to administer the tool in an older population, and we strongly recommend the use of this assessment tool in clinical and research practice.

## Figures and Tables

**Table 1 healthcare-11-01325-t001:** Translation divergences and final solutions chosen by the researcher.

Original	Differences	Solutions	Translation
(a) Item	(1) Parametri (2) Criteri (3) Item	Sono molte le parole in cui si può tradurre il vocabolo “item” in italiano. È stata anche presa in considerazione la possibilità di utilizzare la stessa parola “item” senza tradurla perché ampiamente conosciuta ed utilizzata dagli operatori sanitari italiani	Si è scelto il vocabolo “Criteri” perché più breve e più adatto in quanto il vocabolo “parametri” richiamava un concetto più matematico quantitativo che qualitativo. Si è scartata l’opzione “item” in quanto per quanto diffuso resta un termine straniero.
(b) Measure/s	(1) misurazioni (2) rilevamenti (3) criteri	Il vocabolo “measures” e “measure” viene utilizzato diverse volte nel manuale d’uso e ogni volta assume significati leggermente diversi. In italiano “misurare”, “misure” e “misurazioni” si attribuiscono esclusivamente al quantificare una grandezza.	“Measure/s” è stato tradotto, a seconda dei casi con: “rilevamenti” “valutazioni”, “misurazioni”, “criteri”
(c) User	(1) utente (2) operatore	Il vocabolo user appare solo due volte: nella dicitura “user guide and manual” e “reliability is a function of the user.	Si è scelto di tradurre la stessa parola “user” come utente in relazione alla dicitura “guida utente e manuale d’uso” e come operatore quando si riferisce a colui che utilizza lo strumento e ne determina l’affidabilità.
(d) observations	(1) osservazioni (2) valutazione	“observations” traduce con “osservazioni” che in italiano significa il prodotto dell’osservare così come anche il “fare delle considerazioni”	Si è scelto di optare per “osservazioni” nella maggior parte della traduzione. È stato però scelto di tradurre con “valutazione” quando il contesto lo riteneva più opportuno. Es: “ogni esame o valutazione”
(e) neutral calcaneal stance position	(1) calcagno in appoggio neutro di sottoastragalica (NCSP)	“Neutral calcaneal stance position” viene utilizzato in Italia con l’acronico di NCSP che lo differenzia da RCSP (relaxed calcaneal stance position). Essendo un termine utilizzato nell’ambito clinico dai podologi italiani nella sua forma inglese, non è stato semplice tradurlo.	Si è deciso di tradurre con “appoggio calcaneare in neutra di sottoastragalica” aggiungendo accanto l’acronimo NCSP per sottolineare a cosa si riferisce il termine “neutro”

**Table 2 healthcare-11-01325-t002:** Description of items of the Italian version of the FPI6.

Number of Item	Name of Item	Description	Clinical Note	Score
1	Palpazione della testa dell’astragalo (Palpazione per congruenza talo- navicolare)	Questo è l’unico criterio che assegna un punteggio in base alla palpazione invece che all’osservazione. La testa dell’astragalo è palpata in sede mediale e laterale nell’aspetto anteriore della caviglia, secondo il metodo standard descritto varie volte da Root, Elveru e molti altri. I punteggi vengono assegnati per l’osservazione della posizione in seguito descritta.	Questo non è un tentativo di determinare la cosiddetta posizione neutra di sottoastragalica. Per ottenere il punteggio utile nel test del FPI non l’articolazione nella posizione dove la testa dell’astragalo ha la massima congruenza con il navicolare. Per il FPI viene solamente palpata nella sua posizione rilassata e viene riportata la posizione della testa dell’astragalo. In alcuni casi può tornare utile muovere il piede in inversione ed eversione mentre avviene la palpazione quando risulta difficile assegnare un punteggio di 1 e 2 o –1 e −2.	−2 Testa astragalo Palpabile in Sede laterale/non palpabile −1 Testa astragalo palpabile in sede laterale/poco palpabile in sede mediale 0 Testa astragalo ugualmente palpabile in sede laterale e mediale 1 Testa astragalo poco palpabile in sede laterale/palpabile in sede mediale 2 Testa astragalo non palpabile in sede laterale/palpabile in sede mediale
2	Curvatura Sopra e sotto malleolare (Osservazione e confronto delle curve sopra e sotto i malleoli peroneali)	Nel piede neutro è risultato che le curve dovrebbero essere approssimativamente uguali. Nel piede pronato la curva sotto al malleolo sarà più acuta della curva sopra a causa dell’abduzione del piede e dell’eversione del calcagno. Succede il contrario nel piede supinato.	Nota clinica 1: per stimare la curvatura malleolare, può essere utile utilizzare un righello come riferimento. Questo può essere una squadra, un righello o anche una penna, in base alla disponibilità. Nota clinica 2: Laddove l’edema o l’obesità oscurano la curvatura, questa misura dovrebbe essere valutata a zero o rimossa dalla valutazione e indicata come tale.	−2 Curva sotto al malleolo piatta o convessa −1 Curva sotto al malleolo concava ma più piatta/più bassa della curva sopra al malleoli 0 Entrambe le curve sono uguali 1 Curva sotto al malleolo più concava della curva sopra al malleolo 2 Curva sotto al malleolo molto più concava rispetto alla curva sopra al malleolo
3	Posizione del calcagno sul piano frontale (Inversione/eversione del calcagno)	Questo è un equivalente osservazionale delle tecniche spesso impiegate per quantificare la posizione di appoggio calcaneare rilassata e in neutra di sottoastragalica. Con il paziente in piedi, in posizione rilassata, l’aspetto posteriore del calcagno viene visualizzato con l’osservatore in linea con l’asse lungo del piede. Le misurazioni angolari non sono richieste nel FPI. Il piede viene valutato in base alla visione sul piano frontale.	None	−2 Maggiore di 5° circa di inversione (varo) −1 Tra verticale e circa 5° di inversione (varo) 0 Verticale 1 Tra verticale e circa 5° di eversione (valgo) 2 Più di circa 5° di eversione (valgo)
4	Sporgenza nella regione dell’articola- zione talo- navicolare (TNJ)	Nel piede neutro l’area cutanea immediatamente superficiale al TNJ sarà piatta. Il TNJ diventa più prominente se la testa dell’astragalo è addotta nella pronazione del retropiede. La sporgenza in quest’area è quindi associata a un piede pronato. Nel piede supinato quest’area può essere rientrata.	Una sporgenza nella zona del TNJ è comunemente presente nei piedi pronati. Tuttavia, una vera convessità dell’area è solitamente visibile solo in posture marcatamente supinate. A meno che non vi sia un riscontro ben percettibile, assegnare un punteggio negativo a quest’osservazione dovrebbe essere fatto con cautela	−2 Area del TNJ molto concava −1 Area del TNJ leggermente ma senza dubbio concava 0 Area del TNJ piatta 1 Area del TNJ appena sporgente 2 Area del TNJ notevolmente sporgente
5	Altezza e congruenza dell’arco longitudinale mediale	Mentre l’altezza dell’arco è un forte indicatore della funzione del piede, anche la forma dell’arco può essere altrettanto importante. In un piede neutro la curvatura dell’arco dovrebbe essere relativamente uniforme, simile a un segmento della circonferenza di un cerchio. Quando un piede è supinato, la curva del MLA diventa più acuta all’estremità posteriore dell’arco. Nel piede eccessivamente pronato l’MLA si appiattisce al centro quando il mediotarso e le articolazioni di Lisfranc si aprono.	mentre la semplice altezza dell’arco sarà di solito la misura che più in fretta apparirà all’osservatore, tra le due componenti, la congruenza dell’arco è probabilmente più impercettibile ed informativa. Un’osservazione attenta della congruenza dell’arco dovrebbe essere l’elemento principale di questa valutazione, vedendo quindi la misurazione dell’altezza dell’arco un fattore secondario.	−2 Arco alto e acuto, angolato verso l’estremità posteriore dell’arco mediale −1 Arco moderatamente alto e leggermente acuto posteriormente 0 Altezza dell’arco normale e congruenza della curva 1 Arco abbassato appiattito nella porzione centrale 2 Arco molto basso con sporgenza acuta nella porzione centrale—arco totalmente a contatto con il suolo
6	Abduzione/adduzione dell’avampiede sul retropiede.	Se visto direttamente da dietro e in linea con l’asse lungo del tallone (non l’asse lungo dell’intero piede), il piede neutro consentirà all’osservatore di vedere l’avampiede in modo uguale sui lati mediale e laterale. Nel piede supinato l’avampiede si adduce sul retropiede, risultando visibile una parte maggiore dell’avampiede sul lato mediale. Al contrario, la pronazione del piede provoca l’abduzione dell’avampiede con conseguente maggiore visibilità dell’avampiede sul lato laterale.	questa misurazione dovrebbe essere trattata con attenzione nel caso ci fosse una deformità fissa in adduzione dell’avampiede rispetto al retropiede fuori carico. Solitamente è possibile per l’osservatore vedere le dita sollevando di poco il proprio angolo di visuale. Se le dita sono oscurate da altre strutture, l’articolazione MTF (metatarsofalangea) o strutture più prossimali possono essere usate come guida.	−2 Nessuna delle dita visibile lateralmente. Dita mediali chiaramente visibili −1 Dita mediali maggiormente visibili di quelle laterali 0 Dita mediali e laterali visibili in egual misura 1 Dita laterali più visibili di quelle mediali 2 Nessuna delle dita visibile medialmente. Dita laterali chiaramente visibili

**Table 3 healthcare-11-01325-t003:** Demographic characteristics of the population.

Population	
	**Population 68**
Gender F n (%)	37 (54)
Age mean ± SD	47.1 ± 15.5
Weight Kg mean ± SD	73.1 ± 16.4
Height m mean ± SD	1.7 ± 0.09
BMI mean ± SD	25.3 ± 5.3

**Table 4 healthcare-11-01325-t004:** Inter and intra rater reliability.

Reliability Inter Operator				
	Test (Mean ± SD)	Re-Test Test (Mean ± SD)	ICC	IC 95%
Item 1	0.83 ± 0.986	0.97 ± 0.809	0.83	0.65–0.92
Item 2	0.63 ± 0.850	0.67 ± 0.884	0.88	0.75–0.94
Item 3	0.50 ± 1.009	0.47 ± 1.008	0.90	0.78–0.95
Item 4	0.60 ± 0.814	0.83 ± 0.791	0.74	0.45–0.87
Item 5	0.60 ± 0.855	0.67 ± 0.844	0.95	0.90–0.98
Item 6	0.57 ± 0.858	0.63 ± 0.928	0.88	0.75–0.94
TOT FPI 6	3.73 ± 4.586	4.23 ± 4.272	0.96	0.92–0.98
**Reliability Intra Operator**				
	**Operator 1** **(Mean ± DS)**	**Operator 2** **(Mean ± SD)**		
Item 1	0.83 ± 0.986	0.50 ± 1.009	0.75	0.49–0.88
Item 2	0.63 ± 0.850	0.83 ± 0.913	0.83	0.65–0.92
Item 3	0.50 ± 1.009	0.63 ± 0.928	0.83	0.65–0.92
Item 4	0.60 ± 0.814	0.27 ± 0.691	0.66	0.29–0.84
Item 5	0.60 ± 0.855	0.63 ± 0.928	0.89	0.77–0.95
Item 6	0.57 ± 0.858	0.70 ± 0.915	0.85	0.70–0.93
TOT FPI 6	3.73 ± 4.586	3.57 ± 4.141	0.97	0.92–0.98

**Table 5 healthcare-11-01325-t005:** Item-total statistics.

	Scale Mean If Item Deleted	Scale Variance If Item Deleted	Corrected Item-Total Correlation	Squared Multiple Correlation	Cronbach’s Alpha If Item Deleted
ITEM1	2.8824	12.523	0.759	0.607	0.877
ITEM2	3.1765	12.565	0.779	0.644	0.874
ITEM3	3.2647	12.198	0.808	0.704	0.869
ITEM4	3.1176	13.807	0.785	0.664	0.877
ITEM5	3.2794	14.592	0.552	0.399	0.906
ITEM6	3.0294	13.402	0.708	0.578	0.885

**Table 6 healthcare-11-01325-t006:** Correlation weight/height/BMI/FPI.

	TOT FPI 6
Weight Kg	−0.219
Height cm	−0.215
BMI	−0.155

## Data Availability

The data that supports the findings of this study are available from corresponding upon reasonable request.

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
