# Peer review of "Translation, Cross-Cultural Adaptation, and Validation of the Foot Posture Index (FPI-6)—Italian Version"

_healthcare, 2023, doi:10.3390/healthcare11091325_

Round 1

Reviewer 1 Report

The authors presented an interesting idea of verifying the Foot Posture Index for Italian researchers. The article submitted for evaluation raises some questions:

1. Why was the group of volunteers selected randomly (number of volunteers, gender, age)? It does not make it possible to compare the results between the subgroups, because they are too small. Maybe we should have chosen a different group? Please explain.

2. It is arguable to remove the age limit for subjects (lines 194-196), as opposed to the original FPI text (lines 191-193). No explanation was given for this decision. The term "However, in this study scholars decided to include subjects from 18 years of age with no maximum age limit because the elderly are part of the population to which the Foot Posture Index was addressed" is not convincing. Please complete the text with the arguments of the scholars cited by the authors so that it can be verified.

Reference number 11 has been duplicated in number 12. This error is also visible in the text of the article. Duplicate references 11 and 12 are cited as two independent references.

Articles included in the literature do not have a DOI identifier, which is now a standard in scientific periodicals.

Author Response

Dear reviewer, I really apologize for the delay. We appreciate the opportunity to resubmit our article entitled “Translation, cross-cultural adaptation, and validation of the Foot Posture Index (FPI-6) – Italian version.” We would like to thank the referees for the careful and constructive reviews. We have made corresponding changes directly to the manuscript where appropriate with changes tracked. The revised version of our manuscript accompanies this letter. All comments by the reviewer have been addressed. Based on his/her comments, we have made changes to the manuscript, which are detailed below.

Reviewer Comment

Response

Line #

Reviewer #1

Why was the group of volunteers selected randomly (number of volunteers, gender, age)? It does not make it possible to compare the results between the subgroups, because they are too small. Maybe we should have chosen a different group? Please explain.

Thank you for your comment. We decided to include a sample size of 68 participants, the lack of subgroup analysis has been added to limitation

230, 231

it is arguable to remove the age limit for subjects (lines 194-196), as opposed to the original FPI text (lines 191-193). No explanation was given for this decision. The term "However, in this study scholars decided to include subjects from 18 years of age with no maximum age limit because the elderly are part of the population to which the Foot Posture Index was addressed" is not convincing. Please complete the text with the arguments of the scholars cited by the authors so that it can be verified.

This point has been clarified

194-196

Reference number 11 has been duplicated in number 12. This error is also visible in the text of the article. Duplicate references 11 and 12 are cited as two independent references.

The reference has been corrected

Articles included in the literature do not have a DOI identifier, which is now a standard in scientific periodicals.

DOI have been added

Best regards,

Giovanni Galeoto

Reviewer 2 Report

This is a concise and solid study about  to translate the Foot Posture Index 6 into Italian and assess its inter and intra-observer reliability. The paper is very well presented. I am in favour of its publication. I missed reference in the line 132, regarding the description of the classifications of the statistical analysis. 

Author Response

Dear reviewer, I really apologize for the delay. 

We appreciate the opportunity to resubmit our article entitled “Translation, cross-cultural adaptation, and validation of the Foot Posture Index (FPI-6) – Italian version.” We would like to thank the referees for the careful and constructive reviews. We have made corresponding changes directly to the manuscript where appropriate with changes tracked. The revised version of our manuscript accompanies this letter. All comments by the reviewer have been addressed. Based on his/her comments, we have made changes to the manuscript, which are detailed below.

Reviewer #2

This is a concise and solid study about  to translate the Foot Posture Index 6 into Italian and assess its inter and intra-observer reliability. The paper is very well presented. I am in favour of its publication. I missed reference in the line 132, regarding the description of the classifications of the statistical analysis.

Reference has been added

132

Best regards,

Giovanni Galeoto

Reviewer 3 Report

The aim of this study was to translate, culturally  adapt and measure the psychometric properties of the Italian version of the Foot Posture Index.

L118. On intra-observer reproducibility, Who and When did second assessment performed?

L132. Add the citation.

L169. Please write your summary on based on your results at the first paragraph of discussion. Please write on academic writing.

L228. please write your strength, limitation and clinical implication clearly based on your results compared to previous studies.

Author Response

Dear reviewer, I really apologize for the delay. We appreciate the opportunity to resubmit our article entitled “Translation, cross-cultural adaptation, and validation of the Foot Posture Index (FPI-6) – Italian version.” We would like to thank the referees for the careful and constructive reviews. We have made corresponding changes directly to the manuscript where appropriate with changes tracked. The revised version of our manuscript accompanies this letter. All comments by the reviewer have been addressed. Based on his/her comments, we have made changes to the manuscript, which are detailed below.

Reviewer #3

L118. On intra-observer reproducibility, Who and When did second assessment performed?

The detail has been added

118,119

L132. Add the citation.

Citation has been added

132

L169. Please write your summary on based on your results at the first paragraph of discussion.

Comment addressed

229-231

L228. please write your strength, limitation and clinical implication clearly based on your results compared to previous studies.

These points have been added to conclusion

232-239

Best regards,

Giovanni Galeoto

Reviewer 4 Report

This paper reports the validation of the foot posture index in terms of cross-cultural adaptation, especially in the context of the Italian population. The paper is interesting and could be utilized by medical practitioners to assess the condition of patients in advance to avoid any lower limb injuries. I have the following comments/concerns about the submitted manuscript that could improve its overall quality and readability.

1.     The authors need to provide the main contribution and novelty of their proposed technique over the already reported in the literature.

2.     The abstract has many headings embedded, which should be removed, for the sake of clarity. Line 13 on page 1 in the abstract is not easy to understand.

3.     In section 1, the authors have provided a literature review of the problem. However, this section is divided into many short paragraphs which is not consistent with the journal’s guidelines.

4.     The authors should comment on how Cronbach’s alpha coefficient was calculated.

5.     The authors should also discuss the “Alpha if Deleted” evaluation in detail.

6.     Why only two examiners (medical practitioners) were chosen for this study? I believe this is a very small sample size to make a meaningful conclusion.

7.     The authors should also discuss the standards (ICC 0.97) and (ICC 0.96) in the text before using them for validation.

8.     The conclusion section must be rewritten to provide the significant improvements achieved based on the validation techniques.

9.     More recent references regarding plantar pressure should be included. For example:

A.M. Butt, H. Alsaffar, M. Alshareef, K. K. Qureshi, AI Prediction of Brain Signals for Human Gait Using BCI Device and FBG Based Sensorial Platform for Plantar Pressure Measurements. Sensors, 3085,22, 2022

Park, J.; Kim, M.; Hong, I.; Kim, T.; Lee, E.; Kim, E.-a.; Ryu, J.-K.; Jo, Y.; Koo, J.; Han, S.; Koh, J.-s.; Kang, D. Foot Plantar Pressure Measurement System Using Highly Sensitive Crack-Based Sensor Sensors 2019, 19, 5504.

1.  There are many typos in the manuscript that need to be fixed.

Author Response

Dear reviewer, I really apologize for the delay. We appreciate the opportunity to resubmit our article entitled “Translation, cross-cultural adaptation, and validation of the Foot Posture Index (FPI-6) – Italian version.” We would like to thank the referees for the careful and constructive reviews. We have made corresponding changes directly to the manuscript where appropriate with changes tracked. The revised version of our manuscript accompanies this letter. All comments by the reviewer have been addressed. Based on his/her comments, we have made changes to the manuscript, which are detailed below.

Reviewer #4

1.     The authors need to provide the main contribution and novelty of their proposed technique over the already reported in the literature.

These points have been added to conclusion

232-239

2.     The abstract has many headings embedded, which should be removed, for the sake of clarity. Line 13 on page 1 in the abstract is not easy to understand.

Abstract has been corrected according to reviewer’s comments

10-20

3.     In section 1, the authors have provided a literature review of the problem. However, this section is divided into many short paragraphs which is not consistent with the journal’s guidelines.

Subparagraphs has been removed

introduction

4.     The authors should comment on how Cronbach’s alpha coefficient was calculated.

Description has bee added

129-130

5.     The authors should also discuss the “Alpha if Deleted” evaluation in detail.

Description has bee added

159-161

6.     Why only two examiners (medical practitioners) were chosen for this study? I believe this is a very small sample size to make a meaningful conclusion.

International guidelines recommend to raters to calculate inter-rater reliability. If you refer to the population to which the assessment tool was administered, it was composed by 68 individuals

7.     The authors should also discuss the standards (ICC 0.97) and (ICC 0.96) in the text before using them for validation.

The description has been given in the methods section

8.     The conclusion section must be rewritten to provide the significant improvements achieved based on the validation techniques.

Conclusion has been rewritten

9.     More recent references regarding plantar pressure should be included. For example:

A.M. Butt, H. Alsaffar, M. Alshareef, K. K. Qureshi, AI Prediction of Brain Signals for Human Gait Using BCI Device and FBG Based Sensorial Platform for Plantar Pressure Measurements. Sensors, 3085,22, 2022

Park, J.; Kim, M.; Hong, I.; Kim, T.; Lee, E.; Kim, E.-a.; Ryu, J.-K.; Jo, Y.; Koo, J.; Han, S.; Koh, J.-s.; Kang, D. Foot Plantar Pressure Measurement System Using Highly Sensitive Crack-Based Sensor Sensors 2019, 19, 5504.

References have been added

1.  There are many typos in the manuscript that need to be fixed.

Typos has been corrected

229-236

Best regards,

Giovanni Galeoto

Round 2

Reviewer 3 Report

Thank you for the revision.

No comments

Author Response

Date: April 18, 2023

Dear Editor,

We appreciate the opportunity to resubmit our article entitled “Translation, cross-cultural adaptation, and validation of the Foot Posture Index (FPI-6) – Italian version.” We would like to thank the referees for the careful and constructive reviews. We have made corresponding changes directly to the manuscript where appropriate with changes tracked. The revised version of our manuscript accompanies this letter. Based on reviewr's comments, we have asked a native English speaker to correct typos and any remaining grammatical issue.

We hope that the new version of our manuscript is acceptable for publication.

Best regards,

Giovanni Galeoto

Reviewer 4 Report

The authors should reduce the number of paragraphs in the introduction section. This can be done by combining the short paragraphs.

Please fix the typos still found in the text by letting a native English language speaker go through the text and provide you with the correction. 

Author Response

(The authors gave the same response as above.)
